# Genome-Wide Identification of the GPAT Family in *Medicago sativa* L. and Expression Profiling Under Abiotic Stress

**DOI:** 10.3390/plants13233392

**Published:** 2024-12-03

**Authors:** Jianzhi Ma, Mingyang Du, Huiyan Xiong, Ruijun Duan

**Affiliations:** 1College of Eco-Environmental Engineering, Qinghai University, Xining 810016, China; 18194578185@163.com (J.M.); d2501457943@outlook.com (M.D.); 2Academy of Agriculture and Forestry Sciences, Qinghai University, Xining 810016, China; 3College of Agriculture and Animal Husbandry, Qinghai University, Xining 810016, China; huiyanxqh@163.com

**Keywords:** *Medicago sativa* L., glycerol-3-phosphate acyltransferase (GPAT), gene family, abiotic stress, expression analysis

## Abstract

Glycerol-3-phosphate acyltransferase (GPAT), as a rate-limiting enzyme engaged in lipid synthesis pathways, exerts an important role in plant growth and development as well as environmental adaptation throughout diverse growth stages. Alfalfa (*Medicago sativa* L.) is one of the most significant leguminous forages globally; however, its growth process is frequently exposed to environmental stress, giving rise to issues such as impeded growth and decreased yield. At present, the comprehension of the *GPAT* genes in alfalfa and their reactions to abiotic stresses is conspicuously deficient. This study identified 15 GPATs from the genome of “Zhongmu No. 1” alfalfa, which were phylogenetically categorized into three major groups (Groups I ~ III). Furthermore, Group III is further subdivided into three distinct subgroups. MsGPATs belonging to the same subfamily exhibited similar protein conserved motifs and gene structural characteristics, in which groups with simple conserved motifs had more complex gene structures. A multitude of regulatory *cis*-elements pertinent to hormones and responses to environmental stress were detected in their promoter regions. In addition, a spatial–temporal expression analysis showed that *MsGPATs* have significant tissue specificity. Furthermore, the transcriptomic analysis of ABA treatment and the qRT-PCR results under drought, salt, and cold stress demonstrated that the majority of *MsGPATs* respond to abiotic stress with pronounced timely characteristics. It was also ascertained that these *GPAT* genes might assume a crucial role in salt and drought stress. This research can further constitute a fundamental basis for the exploration of the alfalfa GPAT family, the screening of key GPATs, and the investigation of their functionalities.

## 1. Introduction

Plant lipids constitute a diverse array of organic compounds that exert critical functions in the structure and operation of plant cells. They mainly encompass triglycerides, phospholipids, glycolipids, and sterols. These compounds are essential components of cell membranes, storage structures, extracellular protective layers, and signaling molecules [1]. Cuticular waxes and cutin play crucial roles as essential lipids that form a protective barrier for plants against biotic and abiotic stress, while also regulating the transportation of water and solutes [2]. Glycerol-3-phosphate acyltransferases (GPATs) are crucial enzymes facilitate glycerol lipid biosynthesis by transferring acyl groups from acyl-CoA(coenzyme A) or acyl-ACP (Acyl carrier protein) to the sn-1 position of G3P (3-phosphoglycerol), resulting in the formation of LPA (lysophosphatidic acid), a key intermediate in the synthesis of extracellular lipid polyesters, as well as storage and membrane lipids; these GPATs belong to the sn-1 type [3]. In contrast, other GPATs exhibit phosphatase activity that facilitates the formation of 2-MAG (monoacylglycerol), which does not yield LPA as a primary product; these GPATs are classified as sn-2 type [4,5]. The sn-2 type GPAT specifically promotes the acylation of G3P at the sn-2 site, manifesting a propensity for oxidized lipid acyls. This enzyme assumes a pivotal role in safeguarding the stability of plant membrane lipids and is implicated in the biosynthesis of diverse lipids, encompassing cuticular waxes and cutin, thereby engaging in the signal transduction pathways associated with abiotic stress responses.

Currently, the GPAT family is well studied in *Arabidopsis thaliana*, with 10 *AtGPAT*s characterized, including AtGPAT1-AtGPAT9 and ATS1 [6]. ATS1 and AtGPAT9 are of the sn-1 type. In a previous study, these GPATs were classified into three distinct clades: Group I (ATS1), Group II (AtGPAT9), and Group III (AtGPAT1-AtGPAT8). Group III can also be divided into three sub-branches (AtGPAT1-*3*, AtGPAT5/AtGPAT7, and AtGPAT4/AtGPAT6/AtGPAT8) [7]. Studies have reported that GPAT proteins belonging to different branches have different biological functions. For instance, the up-regulation of *ATS1* enhances the accumulation of unsaturated lipids in the plasma membrane and mitigates the degree of phase transition of the plasma membrane under low-temperature circumstances, thereby augmenting the plant’s tolerance to cold stress [8]. *AtGPAT9* is critical for the biosynthesis of TAG (triacylglycerols) in Arabidopsis, and its expression level is reduced; there is a notable decrease in the accumulation of TAGs within developing seeds, which results in severe phenotypic consequences related to gametophyte development [9]. AtGPAT1-8 are a group of acyltransferases that exhibit characteristics typical of the sn-2 type, which is significant in lipid metabolism. These enzymes play a crucial role in the synthesis of cutin and suberin, two important biopolymers found in plant tissues [10,11]. AtGPAT1, AtGPAT2, and AtGPAT3 are implicated in the development of plant pollen and anthers and are predominantly expressed in flowers and siliques to contribute to the structural and functional integrity of pollen grains [12]. In the *Atgpat1* mutant, the fibrous material and vacuoles in anther compartments are reduced, and pollen grains collapse [13]. AtGPAT4, AtGPAT6, and AtGPAT8 are biofunctional acyltransferase/phosphatase enzymes that play significant roles in various metabolic pathways within plants, with AtGPAT4 and AtGPAT6 related to cutin synthesis and pollen development. The mutant of *AtGPAT6* has abnormal flower organ morphology, and seed setting rate and cutin content were decreased in flowers [14]. AtGPAT4 is functionally redundant with AtGPAT8, and the *Atgpat4*/*Atgpat8* double mutant has a severe blockage of cutin synthesis, increased cuticle permeability, and accelerated water loss [11,15]. Additionally, AtGPAT5 and AtGPAT7 are involved in root and seed suberin synthesis. In the *Atgpat5* mutant, the content of root suberin, ω-OH fatty acid, and DCA (dicarboxylic acid) decreased by 50% and 80% compared with the wild type, with a significant increase in salt permeability and reduced salt tolerance [16].

Simultaneously, numerous GPAT studies related to abiotic stresses have been conducted in non-model plants. The overexpression of the tomato *LeGPAT* in *Arabidopsis thaliana* elevates the level of PG (Phosphatidylglycerol), and the plants exhibit a superior ability to adapt to cold circumstances [17]. Overexpression of *Suaeda salsa SsGPAT* in Arabidopsis led to increased plant salt tolerance through the accumulation of unsaturated fatty acid in PGs [18]. The overexpression of the *AmGPAT* derived from *Ammopiptanthus mongolicus* in Arabidopsis can markedly augment the content of unsaturated fatty acids within plant tissues, thereby facilitating the plant to maintain the fluidity and stability of its membranes under low-temperature stress [19]. Under low-temperature stress, *Paeonia lactiflora PlGPAT* was highly expressed and increased the saturated lipid content of leaf vesicle membranes, thereby enhancing tolerance [20]. At low temperatures, the expression level of *AhGPAT9* in peanuts is significantly increased, which helps to improve the plant’s cold resistance [21]. These studies demonstrate a significant correlation between *GPAT* and plant fertility, seed oil content, and abiotic stress.

Alfalfa (*Medicago sativa* L.), a perennial flowering plant belonging to the legume family, is extensively distributed worldwide. Renowned for its high yield, superior grass quality, and richness in crude protein and other nutrients, it exhibits a wide range of growth adaptability and strong resistance to adverse conditions, earning it the title “king of grass” [22]. Additionally, alfalfa possesses robust nitrogen-fixing capacity and a well-developed root system, which contribute to the prevention of sandstorms and the protection of the ecological environment [23]. At present, the research on alfalfa variety primarily focus on the characteristics of plants at various growth stages under salt stress and the proteomic analysis of salt stress response proteins [24,25]. With the publication of the genome of the alfalfa varieties “Zhongmu No. 1” and “Xinjiang Daye”, it is beneficial to the subsequent exploration of functional genes [26,27]. Consequently, it is essential to further investigate the function of stress-related genes in modulating the growth, development, and responses of alfalfa to abiotic stresses. GPAT participates in various lipid biosynthesis pathways and is vital for plant growth and resistance to stress. It has been reported that 73 members of the GPAT family were identified based on the tetraploid genome data of “Xinjiang Daye” Alfalfa, which can be classified into three subgroups, with diversity in gene structures, protein conserved motifs, and the expression patterns of *MsGPATs* under saline–alkali stress were analyzed with “Zhongmu No. 1” and salt-sensitive variety “WL323” [28]. Nonetheless, knowledge regarding the GPAT family in the “Zhongmu No. 1” genome and its expression patterns in different tissues and in response to abiotic stresses, including cold, drought, and salinity, remains insufficient. This study involved a comprehensive investigation of the GPAT family, encompassing physicochemical properties, phylogenetic relationships, gene structure, promoter *cis*-elements, and duplication event analysis using the alfalfa genome. Moreover, the *GPAT* expression profile in different tissues of alfalfa and under various abiotic stresses were also analyzed. The results of this research are of great significance in revealing the factors affecting the growth of alfalfa, guiding agricultural production and breeding, as well as laying a foundation for future investigations into the biological functions of alfalfa *GPAT* in response to abiotic stress.

## 2. Results

### 2.1. Identification and Chromosomal Localization of MsGPATs in the M. sativa Genome

A total of 27 GPAT members were identified by Blastp (Protein Sequence Alignment) in the *M. sativa* genome using the 10 AtGPAT proteins from *A. thaliana* as a query sequence. Additionally, 21 GPAT members were retrieved by HMM (hidden Markov model) search to query the alfalfa genome database using the GPAT domain (PF01553). After structural domain validation with NCBI-CDD (https://www.ncbi.nlm.nih.gov/cdd/ (accessed on 6 August 2023)) and SMART (http://smart.embl-heidelberg.de/ (accessed on 6 August 2023)), 15 *GPAT*s were confirmed and distributed across six alfalfa chromosomes. Based on their chromosomal positions, these genes were named *MsGPAT1* to *MsGPAT15*. Chromosomal visualization (Figure 1) showed that the *MsGPAT*s are evenly distributed on six of the eight chromosomes, excluding chromosomes 3 and 6. Chromosomes with two genes: chromosome 2 (*MsGPAT4* and *MsGPAT5*), chromosome 4 (*MsGPAT6* and *MsGPAT7*), and chromosome 8 (*MsGPAT14* and *MsGPAT15*); chromosomes with three genes: chromosome 1 (*MsGPAT1*, *MsGPAT2*, and *MsGPAT3*), chromosome 5 (*MsGPAT8*, *MsGPAT9*, and *MsGPAT10*), and chromosome 7 (*MsGPAT11*, *MsGPAT12*, and *MsGPAT13*); with a pair of tandemly duplicated genes identified on chromosome 1.

For the 15 MsGPAT proteins, their molecular weight (MW), protein size (aa, amino acids), isoelectric point (pI), and grand average of hydropathicity (GRAVY) were determined via the Expasy online service (https://web.expasy.org/protparam/ (accessed on 7 August 2023)) (Table 1). The amino acid lengths of MsGPAT proteins ranged from 247 aa (MsGPAT4) to 1329 aa (MsGPAT14), with an average length of approximately 500 aa, corresponding to molecular weights ranging from 28.93 kDa to 150.73 kDa. The pI values of MsGPAT proteins spanned from 7.85 (MsGPAT2) to 9.80 (MsGPAT8), indicating alkaline proteins. Hydrophilicity analysis showed that 9 of 15 are hydrophobic except MsGPAT4, 5, 8, 13, 14, and 15, which are hydrophilic proteins. Subcellular location predictions revealed that MsGPAT9 and MsGPAT14 were localized in the cell membrane, while the remainder were detected in the endoplasmic reticulum, chloroplast, and mitochondria, suggesting that MsGPAT proteins primarily accumulate and function within organelles.

### 2.2. Phylogenetic and Synteny Analysis of MsGPATs

For the purpose of conducting the classification, a phylogenetic tree was constructed by means of the neighbor-joining (NJ) methodology via MEGA 11.0 software using the sequences of 10 *A. thaliana*, 15 *M. sativa*, and 24 *M. truncatula* GPAT proteins (Figure 2). The MsGPATs were clustered into three groups (Groups I ~ III), similar to their counterparts in *M. truncatula* and *A. thaliana*. MsGPAT8, MsGPAT13, and MsGPAT9 belong to Group I and are closely related to ATS1; members of Group II, including MsGPAT15, MsGPAT4, MsGPAT7, MsGPAT5, and MsGPAT12, exhibit close relationships with AtGPAT9; the remaining seven MsGPATs are categorized within Group III and show strong similarities to AtGPAT1 through AtGPAT8. Group III can be further divided into three sub-branches, in which Group IIIA comprises three MsGPATs (MsGPAT3,10) and three AtGPATs (AtGPAT1,2,3), Group IIIB contains two MsGPATs (MsGPAT1,2) and two AtGPATs (AtGPAT5,7), and Group III C includes three MsGPATs (MsGPAT6,11,14) and three AtGPATs (AtGPAT4,6,8).

To investigate the evolutionary relationships of MsGPATs, synteny analysis of GPATs was conducted across three dicots (Figure 3). A total of 9 out of 15 MsGPAT proteins exhibited collinear relationships with *Arabidopsis thaliana*, 14 with *Medicago truncatula*, and 11 with *Glycine max*. Among these gene pairs, there were 11 pairs in *Arabidopsis thaliana*, 19 pairs in *Medicago truncatula*, and 31 pairs in *Glycine max*. The quantity of orthologous pairs between alfalfa and *Glycine max* as well as *Medicago truncatula* was higher than that between *Medicago sativa* and *Arabidopsis thaliana*, suggesting a greater homology among the GPAT family members of alfalfa compared to those from *Medicago truncatula* and *Glycine max*.

### 2.3. Conserved Motifs and Gene Structures of MsGPATs

A unique clustering of alfalfa GPAT proteins is depicted in Figure 4a. The results showed that the cluster tree was consistent with the phylogenetic tree constructed from the GPAT protein sequences of the three plant species shown in Figure 2. The conserved motifs of the MsGPAT proteins were detected using MEME (Figure 4b). Genes belonging to the same subfamily demonstrate resemblance in protein conserved motifs and gene structure. The motifs of Group I and Group II are relatively straightforward, encompassing one to three motifs, while those of Group III are rather intricate, spanning from 7 to 14 motifs. Among them, conserved Motifs 1, 5, and 12 constitute the typical structural domains of acyltransferases (PlsC). The shared conserved motif of the structural domain within Group III is Motif 1; for MsGPAT4 and MsGPAT15 in Group II, it is Motif 5; and for MsGPAT5, MsGPAT7, and MsGPAT12 in the same group, it is Motif 12. It is noteworthy that proteins in Group I display a low quantity of conserved motifs that remain unclear—likely attributable to their shorter protein sequences.

The gene structure of *MsGPATs* revealed variability of introns and exons across each gene (Figure 4c). Notably, the gene structures of Group I and Group II are relatively straightforward, whereas those in Group III exhibit greater complexity. These observations, coupled with the results regarding conserved motifs, suggest that MsGPATs within the same subfamily share similar structural characteristics. Furthermore, there is an inverse relationship between the composition of conserved motifs and gene structures; specifically, simpler conserved motifs correspond to more complex gene architectures, while more intricate conserved motifs are associated with simpler gene structures.

### 2.4. Cis-Acting Elements in the Promoter Regions of MsGPATs

To acquire profound comprehension regarding the transcriptional regulation and potential functions of *MsGPAT* genes, *cis*-regulatory elements within the 1500 bp upstream region of these genes were displayed using PlantCARE (http://bioinformatics.psb.ugent.be/webtools/plantcare/html/ (accessed on 23 August 2023)). A constellation of elements associated with light response, hormonal signaling, and stress response were ascertained to be relatively copious (Figure 5). Light response elements, particularly GT1 motifs, were prevalent in 90% of the *MsGPAT* upstream regions. Stress-related elements (MBS, ARE, and LTR) were also abundant, with AREs (anaerobic response elements) present in nearly 80% of upstream regions, indicating a potential association with abiotic stress responses. Furthermore, hormone-related *cis*-elements such as ABRE (abscisic acid), P-box (gibberellin), TGACG-motif (MeJA), and AuxRE (auxin) were also identified, suggesting regulation by multiple phytohormones.

### 2.5. Expression of MsGPATs in Different Tissues and Under ABA Treatment

To ascertain the expression status of *MsGPATs* in diverse tissues and under ABA treatment, an expression profile was constructed by employing the publicly accessible RNA-seq data acquired from NCBI (https://www.ncbi.nlm.nih.gov/ (accessed on 17 August 2023)) (Figure 6). Due to the incompleteness of RNA-seq data, *MsGPAT14* and *MsGPAT15* were not identified, resulting in a total of 13 genes available for analysis. Spatial expression profiles revealed significant tissue specificity (Figure 6a, Appendix A), with five genes (*MsGPAT1*, *MsGPAT2*, *MsGPAT6*, *MsGPAT11*, and *MsGPAT12*) exhibiting high expression levels in flowers; in elongating stems, higher expression was observed for *MsGPAT5*, *MsGPAT7*, *MsGPAT10*, and *MsGPAT13*; both *MsGPAT8* and *MsGPAT9* demonstrated elevated expression in leaves; the expression of *MsGPAT4*, *MsGPAT5*, and *MsGPAT13* were relatively high in roots; however, gene expression was not significantly detected in nodules.

Furthermore, the expression levels of *MsGPATs* under ABA treatment exhibited significant timely (Figure 6b, Appendix A). The results indicated that three genes (*MsGPAT5*, *MsGPAT7*, and *MsGPAT9*) demonstrated markedly elevated expression levels at 1 h and 3 h; *MsGPAT1* and *MsGPAT2* displayed high expression levels at 12 h; while *MsGPAT3*, *MsGPAT4*, and *MsGPAT12* showed increased expression at 1 h and 3 h. Additionally, *MsGPAT6*, *MsGPAT8*, *MsGPAT10*, *MsGPAT11*, and *MsGPAT13* exhibited significantly high expression levels at 3 h and 12 h.

### 2.6. Gene Expression Pattern of MsGPATs Under Abiotic Stresses

To elucidate the expression patterns of *MsGPAT*s under abiotic stress more deeply, nine genes from three subfamilies were randomly selected to assess their expression levels under cold, drought, and salt treatments using qRT-PCR (Figure 7). The expression level of *MsGPATs* was also analyzed under abiotic stresses. Under cold stress, *MsGPAT15* was significantly up-regulated at 9 h, whereas seven genes (*MsGPAT5*, *MsGPAT6*, *MsGPAT8*, MsGPAT9, *MsGPAT10*, *MsGPAT11*, and *MsGPAT13*) were consistently down-regulated compared to the control across all treatment durations, indicating a response to cold stress (Figure 7a, Appendix A). In contrast, under drought stress (Figure 7b, Appendix A), the expression level of *MsGPAT5* decreased at 6 h, whereas *MsGPAT10* was up-regulated at all different treatment times, reaching approximately 30-fold the expression of the control group at 9 h. Both *MsGPAT3* and *MsGPAT15* were both up-regulated at 24 h; additionally, *MsGPAT6*, *MsGPAT8*, *MsGPAT9*, *MsGPAT11*, and *MsGPAT13* were up-regulated at 24 h. Furthermore, these genes displayed fluctuating expression patterns with increased treatment time. Under salt stress (Figure 7c, Appendix A), *MsGPAT3* was significantly expressed at 3 h; meanwhile, *MsGPAT5*, *MsGPAT6*, *MsGPAT8*, *MsGPAT9*, *MsGPAT10*, *MsGPAT11*, *MsGPAT13*, and *MsGPAT15* reached peak expression levels at 9 h. The results indicated that the response of *MsGPAT*s to salt stress was primarily observed at 9 h, suggesting that these genes respond positively to salt stress.

## 3. Discussion

Plant lipid synthesis-associated GPATs are widely present in eukaryotic organisms and have an impact on plant growth and adaptation to the environment [29]. Identifying and analyzing the plant GPAT family can provide a deeper understanding of their potential functions. Currently, the GPAT family has been identified in various plants, including 10 genes in Arabidopsis [6], 24 genes in *Medicago truncatula* based on the results of our previous studies, 32 in *Brassica rapa* [30], 20 in maize [31], and 85 in *Gossypium* spp. [32], respectively. In this study, we identified 15 GPAT family members evenly distributed on six chromosomes (Table 1, Figure 1). This represents a reduced number of GPAT family members compared to the above species and the identification of alfalfa 73 GPAT members based on the “Xinjiang Daye” genome [28], because we used the haploid genome of “Zhongmu No. 1”, a heterozygous autotetraploid, which reported a higher-quality chromosome-level genome assembly wherein genes exhibited a low degree of repetition based one set of chromosome [26]. To some extent, this haploid genome avoids the differences between homologous chromosomes and the presence of chimeric sequences of the parental genome, which may enable a more distinct and lucid analysis of the classification and other characteristics of this family, which is more conducive to the subsequent mining and verification of GPATs. Furthermore, significant variations exist in the physicochemical characteristics of these members, such as isoelectric point and acidity/basicity, highlighting the diversity within the gene family.

This study demonstrates that alfalfa exhibits distinct GPAT family collinear relationships with two other leguminous plants. Specifically, among the MsGPATs, 31 pairs of orthologous genes were identified with *Glycine max*, 19 pairs with *Medicago truncatula*, and 11 pairs with *Arabidopsis thaliana*. Alfalfa GPAT is more closely related to *Glycine max* and *Medicago truncatula*, which possess a greater number of orthologous GPAT pairs due to their classification within the legume family. This suggests that GPATs are more conserved among closely related species throughout evolution [33]. Phylogenetic analysis demonstrates that the 15 MsGPATs are classified in accordance with the classification of Arabidopsis GPATs into three subfamilies (Figure 2). It is presumed that closely related proteins share similar biological functions. For instance, MsGPAT8, MsGPAT13, MsGPAT9, and ATS1 are members of Group I. In the *ATS1* mutant, compared with the wild type, the amount of fatty acid accumulation was significantly reduced and phospholipid content decreased by 25%, resulting in slower plant growth and a reduced seed setting rate [34]. This phenotype is correlated with the integrated regulation of lipid metabolism and plant development. Therefore, it is speculated that these two genes could be pertinent in the oil synthesis of seeds in plants. Five proteins (MsGPAT4, MsGPAT5, MsGPAT7, MsGPAT12, and MsGPAT15) along with AtGPAT9 are classified into Group II. AtGPAT9 is crucial for TAG biosynthesis; down-regulation of its expression leads to a reduction in seed oil content and manifests lethal phenotypes in both male and female gametophytes [35]. It is conjectured that these five genes could be associated to the generation of seed TAG. The remaining MsGPATs are classified into Group III, while AtGPAT1-8 are primarily involved in cutin synthesis [36]. Among them, MsGPAT11 and MsGPAT6, which belong to Group IIIA, were manifested a conspicuous up-regulation at 9h in the context of drought and salt stress. It is hypothesized that MsGPAT11 and MsGPAT6 may be involved in cutin synthesis as a response to abiotic stresses. Additionally, most of the MsGPATs are localized in chloroplast, mitochondria, and the endoplasmic reticulum, while a few are found in the cell membrane (Table 1), which aligns with the localization results observed for Gossypium GPAT [32]. Among them, MsGPAT8 and MsGPAT13 are localized in plastids, which is consistent with ATS1 (Group I), while most MsGPATs belonging to the same group are also found in the endoplasmic reticulum. MsGPAT3/MsGPAT10 in Group IIIA are consistently localized to mitochondria alongside AtGPAT1, AtGPAT2, and AtGPAT3. These results indicate that subcellular localization of genes within the same clade is similarly conserved. Based on the phylogenetic tree, conserved motifs, and gene structure analyses of alfalfa MsGPAT proteins (Figure 4), MsGPATs within the same group manifested similar motif types and numbers, while their gene structures displayed analogous characteristics, including comparable exon/intron arrangements, positions, and intron counts. This observation is consistent with findings in the GPAT family of *Medicago truncatula* and *Brassica rapa* [30]. Furthermore, the conserved motifs of MsGPATs belonging to the same subfamily are relatively similar, suggesting that similar subfamilies may perform analogous functions, which could lead to redundancy in gene functions. In contrast, the conserved motifs across different subfamilies exhibit significant differences, indicating that distinct subfamilies may fulfill divergent roles. This finding aligns with evolutionary analyses and suggests that diverse structural features imply varied functions of GPATs. Additionally, we observed an inverse relationship between the composition of conserved motifs and gene structures in MsGPATs; specifically, simpler conserved motifs correspond to more complex gene structures, while more complex conserved motifs are associated with simpler gene structures. This observation is particularly intriguing and may hold special significance for GPAT evolution.

The promoter region of the *MsGPATs* in alfalfa encompasses multiple *cis*-acting elements that are associated with light, hormone response, and stress response. (Figure 5). This indicates that the *MsGPATs* might be implicated in the growth, development, and metabolic processes of plants, and concurrently participates in the biological processes that modulate the responses of plants to various abiotic stresses. Each *MsGPAT* contains multiple light-sensitive elements similar to those found in *Avena sativa* [37]. It has been demonstrated by previous research that light has an impact on the expression of genes associated with lipid synthesis [38], and microalgae accumulate TAG under illuminated conditions [39]. Integrating these findings with promoter analysis leads to the hypothesis that *GPAT* might be implicated in the pathway of light-induced lipid synthesis in plants. Furthermore, spatial–temporal expression analysis of *MsGPATs* revealed that *MsGPAT1*, *MsGPAT2*, *MsGPAT6*, and *MsGPAT11* exhibit particularly high expression levels in flowers (Figure 6a), implying their potential involvement in floral development.

Abscisic acid (ABA) serves as a key regulatory factor for plants in response to environmental and organismal alterations, exerting an extremely vital role in modulating responses to diverse stress conditions [40,41]. ABA also governs the expression of *GPAT*. For instance, it can inhibit the expression of *BnGPAT9* in response to external stimuli [30]. Additionally, *LlaGPAT* is significantly induced by ABA, resulting in up-regulation of its expression [42]. In this study, alfalfa *MsGPAT*s exhibited temporal characteristics in response to ABA treatment. For instance, *MsGPAT1* and *MsGPAT2* demonstrated significant up-regulation, while the expression levels of *MsGPAT7*, *MsGPAT5*, and *MsGPAT9* were markedly down-regulated. Other *MsGPAT*s showed fluctuating expression changes under ABA treatment, i.e., *MsGPAT*s might be involved in ABA-mediated drought stress response. Numerous studies have verified that *GPAT*s are down-regulated under abiotic stress, as observed in sunflower *GPAT* [43], *Ammopiptanthus mongolicus AmGPAT* [19], and *Lepidium latifolium LlaGPAT* [42]. *MsGPAT*s also exhibited timely specificity in response to abiotic stresses. The *MsGPATs* were consistently down-regulated throughout all treatment durations under cold stress, suggesting that these genes might exert a negative regulatory effect in such circumstances. Conversely, the expression levels of *GPAT* from different species increased under other abiotic stresses; for instance, rice *OsGPAT5*, *14*, *18*, *19*, and *24* were significantly up-regulated under both salt and drought stress [44]. In our previous research, barley *HvGPAT5*, *8*, *14*, *17*, *18*, and *HtGPAT14* were up-regulated in the context of salt and drought stress. In the present study, the majority of *MsGPAT*s manifested up-regulation, peaking after 9 h of salt stress treatment. Under drought stress, the expression of *MsGPAT*s fluctuated with each treatment time; except for *MsGPAT5*, which was down-regulated under drought, other *MsGPAT*s were significantly up-regulated at certain times. Notably, *MsGPAT10* was persistently up-regulated throughout all treatment times under drought stress, suggesting its potential pivotal functionality in the response to such conditions.

## 4. Materials and Methods

### 4.1. Identification of GPAT Gene Family

The genome-wide data for alfalfa were procured from (https://figshare.com (accessed on 2 August 2023)). To discern all members of the GPAT gene family in alfalfa, we accessed the sequence and annotation particulars of the Arabidopsis GPAT gene family from the TAIR database (https://www.arabidopsis.org/ (accessed on 2 August 2023)). BLASTp searches (E-value = 0.00001) were performed to identify candidate GPAT proteins based on reference sequences from the Arabidopsis GPAT gene family. Additionally, hidden Markov model (HMM) profiles corresponding to the GPAT domain (PF01553) were downloaded from the Pfam database (http://pfam.xfam.org (accessed on 2 August 2023)) and genes containing these domains were searched for. The potential *M. sativa* GPAT members identified through these two approaches were compiled into a comprehensive dataset. Furthermore, WebCDD-search (https://www.ncbi.nlm.nih.gov/cdd (accessed on 6 August 2023)) and SMART (http://smart.embl.de/ (accessed on 6 August 2023)) were utilized to analyze the domains of candidate GPAT proteins, thereby confirming which genes would be selected for subsequent analysis.

### 4.2. Basic Physicochemical Properties and Chromosomal Location of MsGPATs

We employed the online website ExPASY (https://web.expasy.org/protparam/ (accessed on 7 August 2023)) to analyze the molecular weight (MW), isoelectric point (pI), amino acid count, and average hydrophilicity (GRAVY) of MsGPATs, thereby comprehending their physicochemical properties. The Plant-PLoc server (Plant-PLoc server: sjtu.edu.cn) was utilized to predict the localization of these proteins. Based on the annotation data from the alfalfa genome database, the distribution of the GPATs across chromosomes was analyzed.

### 4.3. Phylogenetic and Synteny Analysis of MsGPAT Proteins

The entire genomic information of *A. thaliana* and *M. truncatula* was retrieved from the NCBI database (https://www.ncbi.nlm.nih.gov/ (accessed on 7 August 2023)). By employing ClustalW technology, the protein sequences of 15 MsGPAT, 10 AtGPAT, and 24 MtGPAT were subjected to multiple sequence alignment. The alignment parameters were set in the multiple comparison mode (with other parameters remaining in the default configuration), and the resultant alignments were utilized to construct a neighbor-joining (NJ) phylogenetic tree, which was generated in MEGA 11.0 through 1000 bootstrap replications. Furthermore, the synteny analysis between MsGPAT proteins and those of *A. thaliana*, *G. max*, and *M. truncatula* GPAT proteins was carried out using JCVI (https://github.com/tanghaibao/jcvi (accessed on 8 August 2023)).

### 4.4. Gene Structure and Conserved Motif Analysis

Employing the GFF (General Feature Format) annotation file of the alfalfa genome, we availed ourselves of the online Gene Structure Display Server (GSDS) (http://gsds.cbi.pku.edu.cn/ (accessed on 8 August 2023)) to engender a visual manifestation of exon–intron structures. The conserved motifs encompassed within MsGPATs were scrutinized using the Multiple Expectation Maximization for Motif Elicitation (MEME) Suite (http://meme-suite.org/ (accessed on 9 August 2023)), with the motif quantity configured to 20.

### 4.5. Analysis of Cis-Acting Elements of MsGPATs

The 1500 bp promoter region located upstream of the transcriptional start site for MsGPAT genes was obtained from the M. sativa genome and analyzed using PlantCARE (https://bioinformatics.psb.ugent.be/webtools/plantcare/html/ (accessed on 23 August 2023)). The results of this analysis were visualized with GSDS online website (http://gsds.gao-lab.org/ (accessed on 23 August 2023)).

### 4.6. Expression Profiling of the MsGPATs in Different Tissues and ABA Treatment

RNA-Seq data for different tissues (elongating stems, post-elongating stems, flowers, leaves, nodules, and roots) and ABA treatments at different times (1 h, 3 h, and 12 h) were downloaded at project numbers PRJNA276155 and PRJNA450305 from the NCBI public database, respectively. The expression levels of *MsGPAT*s in various tissues and under ABA treatments were quantified as fragments per kilobase per million mapped reads (FPKM values). A heatmap representing the expression profile of *MsGPAT*s was generated using R software (version R-4.3.1).

### 4.7. Plant Growth and Abiotic Treatments

The seeds of alfalfa “Zhongmu No. 1” were sterilized and planted in nutrient soil, grown under a light/dark regime of 12 h/12 h, with diurnal/nocturnal temperatures of 30 °C/25 °C and a relative humidity of 65%. The seedlings were then transferred to Hoagland’s nutrient solution for further incubation after one week. Four-week-old alfalfa seedlings were subjected to drought, salt, and cold stress treatments with 15% PEG6000, 250 mM NaCl, and 4 °C, respectively, with seedlings under normal conditions used as controls. Leaf samples were collected after 3 h, 6 h, 9 h, 12 h, 24 h, and 48 h, and store at −80 °C for subsequent quantitative experiments. Three biological replicates were performed for each sample.

### 4.8. Quantitative Real-Time RT-PCR (qRT-PCR) of MsGPAT Genes

The total RNA of alfalfa from each sample was extracted using an RNAprep Pure Plant kit (TaKaRa, Dalian, China), and RNA quality and concentration were detected using a Nano-Drop 2000 UV spectrophotometer. First-strand cDNA was generated from RNA through reverse transcription using the PrimerScript 1st Strand cDNA Synthesis Kit (Tiangen, Beijing, China). After measuring the concentration, the cDNA was uniformly diluted to 100 ng for use as a template in qRT-PCR reactions. Primers for *MsGPATs* were designed using Primer Premier 5.0 software and are detailed in Appendix A, with β-Actin serving as an internal control. qRT-PCR was conducted with SYBR Green (Tiangen, Beijing, China) on a Roche real-time Detection System (Applied Biosystems, Foster City, CA, USA). The thermal cycling conditions included an initial step at 95 °C for 15 min, followed by 40 cycles of denaturation at 95 °C for 10 s, annealing at 60 °C for 20 s, and extension at 72 °C for 32 s. Three biological replicates were performed to calculate the relative gene expression using the 2^−ΔΔCt^ method.

## 5. Conclusions

A total of 15 *MsGPATs* were identified and uniformly distributed across six chromosomes based on the genomic information of alfalfa. This family can be categorized into three clades in accordance with evolutionary relationships. Subcellular localization, conserved motifs, and gene structures display considerable resemblance within each subgroup, but manifested significant disparities between subgroups. The promoter regions of the *MsGPAT*s encompass *cis*-regulatory elements pertinent to stress responses, hormonal modulation, plant tissue morphogenesis, and light perception. Spatial and temporal expression analysis indicated that the *MsGPAT*s showed significant specificity in different tissues. Additionally, qRT-PCR analysis unveiled varying degrees of responsiveness to abiotic stress among the *MsGPAT*s. The majority of *MsGPAT*s were conspicuously up-regulated under drought and salt stress, suggesting their involvement and crucial role in salt and drought stress. Overall, the findings of this research will constitute a significant basis for future studies on the functions of *GPAT* and the molecular mechanisms underlying stress regulation in alfalfa.

## Figures and Tables

**Figure 1 plants-13-03392-f001:**
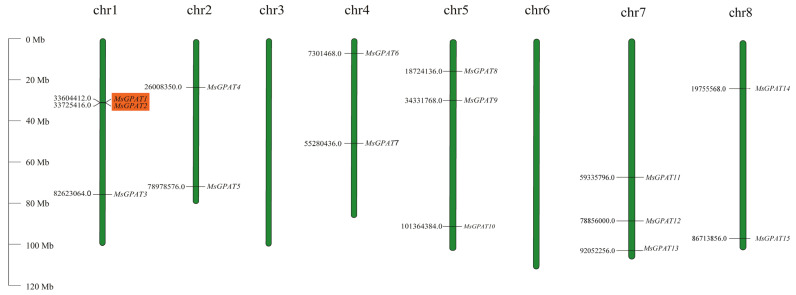
Chromosomal localization of *MsGPAT*s on alfalfa chromosomes. Chromosome numbers are indicated at the top of each respective chromosome. Positional data are displayed on the left side, and the corresponding *MsGP*AT’s name is connected by a short line on the right side. The rectangle marked in orange represent one pair of tandemly duplicated genes. The scale (Mb) represents the length of the chromosome.

**Figure 2 plants-13-03392-f002:**
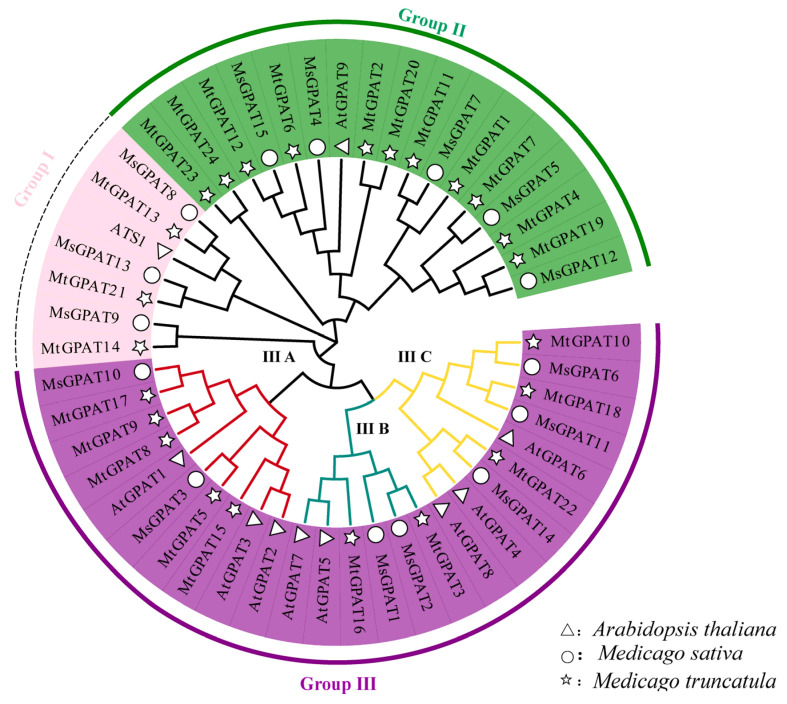
Phylogenetic analysis of the MsGPAT proteins. The GPAT proteins from *M. sativa*, *M. truncatula*, and *A. thaliana* are marked with the circle, pentagram, and triangle, respectively. The 15 MsGPAT proteins can be categorized into three clades (Group I, Group II, and Group III are marked in pink, green, and purple, respectively), and the third cluster can be further divided into three subclades (Group III A, Group III B, and Group III C).

**Figure 3 plants-13-03392-f003:**
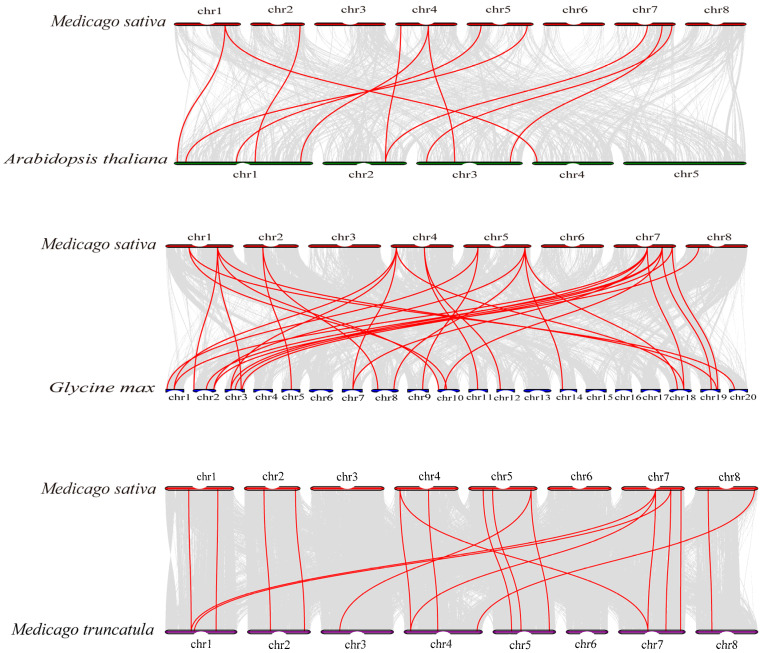
Synteny analysis of *Medicago sativa* GPAT proteins between *Arabidopsis thaliana*, *Medicago truncatula*, and *Glycine max*. Gray lines in the background represent alignment blocks between paired genomes, and red lines indicate syntenic GPAT pairs.

**Figure 4 plants-13-03392-f004:**
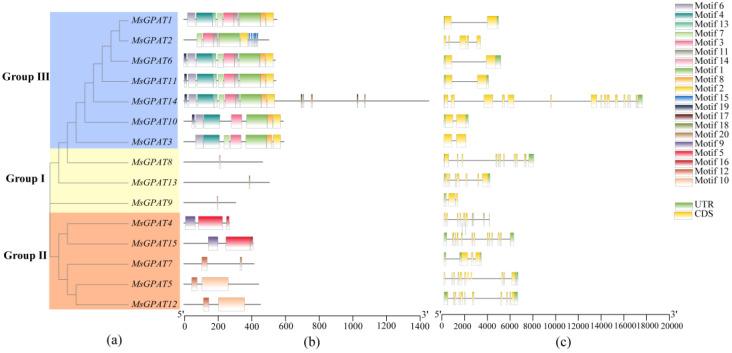
Gene structure of *MsGPAT*s and the conserved motifs in MsGPAT proteins. (**a**) Phylogenetic tree based on the conserved domain of MsGPAT proteins. (**b**) Analysis of conserved motifs in MsGPAT proteins. Colored boxes represent distinct conserved motifs that vary in sequence and size. (**c**) Exon–intron structure of the *MsGPAT*s. The yellow and green rectangles represent coding sequence (CDS) and untranslated region (UTR), respectively, while the black lines denote introns.

**Figure 5 plants-13-03392-f005:**
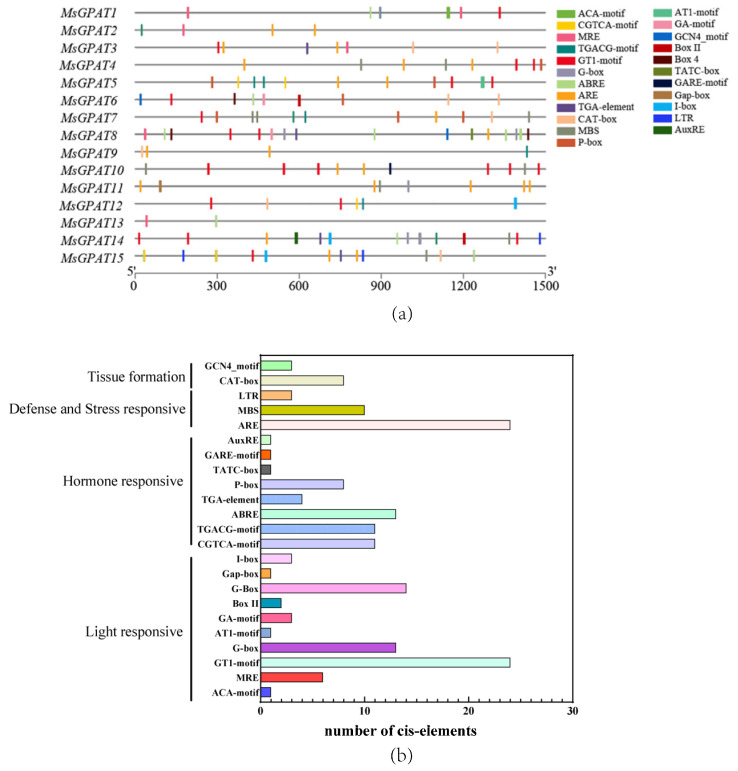
Display of *cis*-acting elements within putative promoters of *MsGPATs*. (**a**) A multitude of *cis*-acting elements were identified within the regulatory regions of each *MsGPAT*, with varying colors and shapes denoting elements. (**b**) The number of *cis*-acting elements.

**Figure 6 plants-13-03392-f006:**
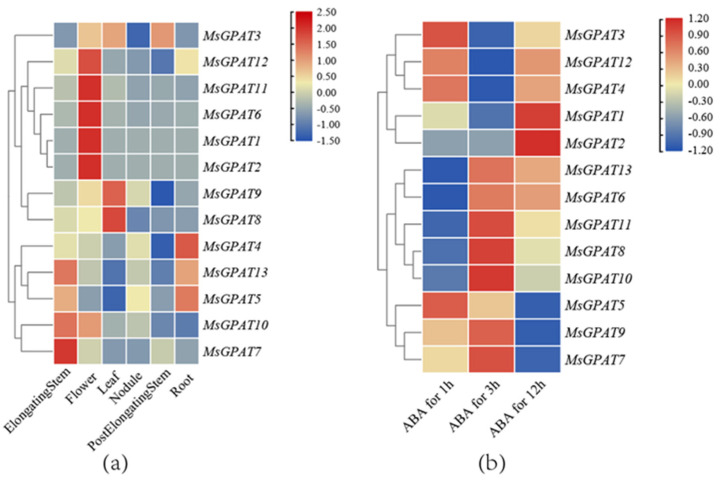
A heat map representation of *MsGPAT* expression between different tissues (**a**) and under ABA treatment (**b**). Red or blue colors represent the difference in expression levels in each sample, according to the color code shown on the right of the heat maps.

**Figure 7 plants-13-03392-f007:**
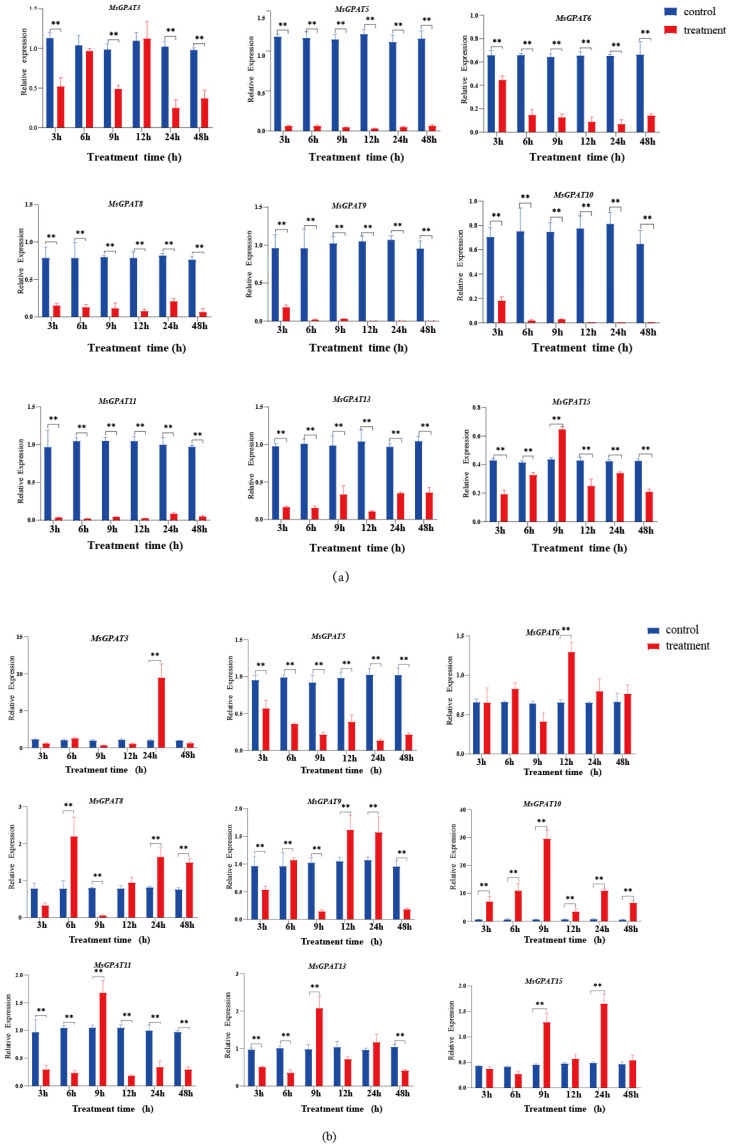
Expression analysis of *MsGPAT*s under abiotic stresses of cold (**a**), drought (**b**), and salt (**c**). X-axis represents the hours of stress treatment. Y-axis represents the expression level of each *MsGPAT*. Data represent the mean ± SE of three replicates. Asterisks represent significant difference at *p* ≤ 0.05 (*) and *p* ≤ 0.01 (**).

**Table 1 plants-13-03392-t001:** Physical and biochemical properties of the *MsGPAT*s.

Gene Name	Gene ID	Chromosome Location	Length of Amino Acid (aa)	Molecular Weight (KDa)	pI	GRAVY	Localization
*MsGPAT1*	MsG0180002144.01.T01	Ch1r	504	55.04199	9.25	0.242	Mitochondrion
*MsGPAT2*	MsG0180002147.01.T01	Ch1r	460	49.86203	7.85	0.119	Endoplasmic reticulum
*MsGPAT3*	MsG0180004776.01.T01	Ch1r	542	61.57195	9.33	0.065	Mitochondrion
*MsGPAT4*	MsG0280008145.01.T02	Ch2r	247	28.93137	9.31	−0.119	Mitochondrion
*MsGPAT5*	MsG0280011060.01.T01	Ch2r	405	47.09177	9.72	−0.201	Mitochondrion
*MsGPAT6*	MsG0480018607.01.T01	Ch4r	495	55.08081	9.34	0.166	Endoplasmic reticulum, mitochondrion
*MsGPAT7*	MsG0480021218.01.T01	Ch4r	380	44.02557	9.28	0.262	Endoplasmic reticulum
*MsGPAT8*	MsG0580025445.01.T01	Ch5r	426	46.53116	8.86	−0.329	Chloroplast
*MsGPAT9*	MsG0580026299.01.T01	Ch5r	281	32.30629	9.8	0.08	Endoplasmic reticulum
*MsGPAT10*	MsG0580029736.01.T01	Ch5r	539	60.86481	9.26	0.17	Mitochondrion
*MsGPAT11*	MsG0780039186.01.T01	Ch7r	498	55.54921	9.27	0.108	Endoplasmic reticulum, mitochondrion
*MsGPAT12*	MsG0780040534.01.T02	Ch7r	415	46.78413	9.43	0.213	Cytomembrane
*MsGPAT13*	MsG0780041529.01.T01	Ch7r	463	52.96296	9.51	−0.425	Chloroplast
*MsGPAT14*	MsG0880043148.01.T01	Ch8r	1329	150.73197	8.6	−0.037	Cytomembrane
*MsGPAT15*	MsG0880047464.01.T01	Ch8r	376	43.25495	9.26	−0.128	Endoplasmic reticulum

## Data Availability

The transcriptome sequencing data of different tissues and ABA treatment are available in the NCBI database (project ID PRJNA276155, PRJNA450305). All methods were carried out in accordance with relevant guidelines and regulations. All experimental studies on plants were complied with relevant institutional, national, and international guidelines and legislation. Further inquiries can be directed to the corresponding authors.

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
