# Peer review of "Genome-Wide Identification of the GPAT Family in Medicago sativa L. and Expression Profiling Under Abiotic Stress"

_plants, 2024, doi:10.3390/plants13233392_

Round 1
Reviewer 1 Report (Previous Reviewer 1)
Comments and Suggestions for Authors
The major issues persist. The authors replied “The identification of MsGPATs gene has repeatability” which is not correct. The publication by Tang et al. (2023) clearly illustrated the distribution of MsGPATs on each chromosome set, demonstrating the diversity of MsGPATs rather than “repeatability”. The chromosomal location, physical and biochemical properties, phylogeny and synteny, gene structure and cis elements of MsGPATs are all analyzed in former studies by Tang et al. (2023). If using the chromosome level ZhongMu No.1 genome for analysis could provide more insights into MsGPATs than using the allele-aware genome as reference, the authors should clearly state the advantage, compare the results and highlight the new findings in the manuscript.
Comments on the Quality of English LanguageThis manuscript is generally readable.
Author Response
Comments and Suggestions for Authors
1
Q1) The major issues persist. The authors replied “The identification of MsGPATs gene has repeatability" which is not correct. The publication by Tang et al. (2023) clearly illustrated the distribution of MsGPATs on each chromosome set, demonstrating the diversity of MSGPATS rather than “repeatability". The chromosomal location, physical and biochemical properties, phylogeny and synteny, gene structure and cis elements of MsGPATs are all analyzed in former studies by Tang et al. (2023). lf using the chromosome level ZhongMu No.1 genome for analysis could provide more insights into MsGPATs than using the allele-aware genome as reference, the authors should clearly state the advantage, compare the results and highlight the new findings in the manuscript. This manuscript is generally readable.
Response: Thank you again for your suggestions. The following are the explanations of our manuscript:
- Alfalfa is known for its ability to hybridize and cross-pollinate between subspecies, resulting in genomes that are highly heterozygous and highly repetitive. The "Zhongmu No. 1" alfalfa genome is a high-quality chromosome-level haploid genome of a heterozygous tetraploid composite, to some extent avoiding the differences between homologous chromosomes and the presence of chimeric sequences of parental genome, which allows for more accurate and comprehensive gene annotation information, benefiting subsequent functional gene mining.Therefore, we selected the "Zhongmu No.1" alfalfa genome as the candidate genome for the study of the GPAT family.
- 2. We have also read the article published by Tang et al. (2023) (doi: 10.11733/j.issn.1007-0435.2023.09.005), which is a worthy reading. In Tang‘s paper, the identification of the GPAT family of alfalfa was based on the tetraploid genomic data of the Chinese local specific variety "Xinjiang Daye" alfalfa. We carefully analyzed the positional characteristics of the GPATs identified in Tang’s article and discovered that, except for a few individual gene deletions, the majority of GPATsexhibit a relatively high degree of repetition among the four sets of chromosomes in their genomes. Given that the identification of the GPAT family in our manuscript is based on the high-quality chromosome level haploid genome of the heterozygous tetraploid "Zhongmu 1", although there are relatively few members of the GPAT family identified, this enables a more distinct and lucid analysis of the classification and other characteristics of this family, and more beneficial to the subsequent mining and verification of GPATgene
- 3. In the article published by Tang et al. (2023), the identification of the GPAT gene family was based on the "Xinjiang Daye" genome, but the subsequent expression pattern analysis under saline-alkali stress was conducted using alfalfa "ZhongmuNo.1" (ZM1) and the foreign-introduced salt-sensitive variety "WL323" as the research objects. In our manuscript, the identification of the GPAT family and the expression analysis of MsGPATsunder abiotic stress were both based on the "zhongmu No.1", and the correspondence and consistency of the results were relatively high.
- In our manuscript, although the GPAT gene family in alfalfa sharing similar characteristics in terms of physicochemical properties,classification and cis-acting elements analysis with the article by Tang et al, we have also summarized some noteworthy unique characteristics: 1) MsGPATs belonging to the same subfamily exhibited similar protein conserved motifs and gene structural characteristics, in which groups with simple conserved motifs having more complex gene structures; 2)The majority of MsGPATs were responsed to ABA, drought, salt, cold stress and have significant timely characteristic.
Above all, it is inferred that these GAPT genes may play an important role in salt and drought stress, which helps to screen and explore the functions of key GPAT genes. Therefore, we maintain that this manuscript possesses certain significance and advantages.
Here is our response to your question. Considering the inconvenience of citing Chinese articles, comparative analysis of GPAT family members identified in two genomes is not suitable for inclusion in the main text.

Reviewer 2 Report (Previous Reviewer 3)
Comments and Suggestions for Authors
It is very incaragias that the authors have resubmitted the article after substantial corrections and improvement. However, even improved, English is still nedded to improve further by an expart.
-As you mentioned, these genes were classified previously as same group as you obtained, why it is necessary again?
L124; make simple, which chromosome contained what?
-whare is description of fig. 4a? if it is duplicate of fig. 2. remove it
-Fig. 4 fig is not legible
-Expression analysis of MsGPATs genes in different tissues and under ABA treatments is already stablished. This needs fresh experiment to confirm this result. This gives significance to the scientific society.
-Discussion section should be revised further with improved Englished commands
-Please check pdf.

Moderate editing of English language required.
Author Response
Responds to the reviewer2’s comments:
lt is very incaragias that the authors have resubmitted the article after substantial corrections and improvement. However, even improved, English is still needed to improve further by an expart.
Q1) As you mentioned, these genes were classified previously as same group as you obtained, why it is necessary again?
Response: Thank you for your suggestion. The reason why we used the existing classification of species GPAT proteins in our manuscript was to analyze the evolutionary relationship of these 15 alfalfa MsGPAT proteins and their homology with other species GPAT proteins. The aim was to illustrate two points: 1) The homology of MsGPAT proteins with other species GPAT proteins; 2. The classification and functional prediction of MsGPAT proteins based on the established classification criteria of Arabidopsis thaliana and Medicago truncatula GPAT proteins. The evolutionary relationships of GPAT proteins and the basis of their classification in this part were referred to the following articles: (doi: 10.4161/psb. 6.11.17777) (doi: 10.1038/boboj. 2013.77) (doi: 10.1111/j. 1365-313X. 2006.02790.x).
Q2) L124; make simple, which chromosome contained what?
Response: Thank you for your advice. We added in lines 137 to141.
Q3) 1. where is description of fig. 4a? 2.if it is duplicate of fig. 2. remove it
Response: 1. Thank you for your advice. We added description of fig. 4a in lines 196 to 198;
- Figure 2 is to study the homology of MsGPAT proteinswith other species, and to analyze the classification and predict the function of MsGPAT proteins according to the established classification criteria and function of other species.Figure 4 illustrates the clustering results for conserved motifs within the MsGPAT family (figure 4b), as well as for gene structure (figure 4c), clustering relationships of the MsGPAT family phylogenetic tree based on the conserved domain of MsGPAT proteins (figure 4a).
Q3) Fig. 4 fig is not legible
Response: 1. Thank you for your advice. We add a new figure in the line 217.
Q4) Expression analysis of MsGPATs genes in different tissues and under ABA treatments is already stablished. This needs fresh experiment to confirm this result. This gives significance to the scientific society.
Response: The suggestions you gave really need to be considered. Based on the fact that alfalfa ‘ZhongMu No.1’ is a variety with the characteristics of salt, cold, drought resistance, the main purpose of our manuscript is to verify the response of MsGPATs to abiotic stresses, so as to provide a basis for the subsequent selection of the key genes for the abiotic stresses resistance. The expression analysis of MsGPAT genes in different tissues and ABA treatments is based on transcriptome data, and the content of this part is mainly to support our manuscript.
Your suggestion does need to be considered. Based on the fact that alfalfa 'Zhongmu 1' is a variety with salt tolerance, cold resistance, and drought resistance characteristics, the main purpose of this article is to verify the response of MsGPAT to abiotic stress and provide a basis for selecting key genes for resistance to abiotic stress in the future. The expression analysis of MsGPAT gene in different tissues and ABA treatment is based on transcriptome data, and this section is mainly to support our manuscript.
Q5) Discussion section should be revised further with improved Englished commands
Response: Thank you for your advice. We checked and modified the English commands that appeared in the discussion section.
Q6) Please check pdf. peer-review-40387296.v1.pdf. Moderate editing of English language required.
Response: Thank you for your suggestion. We have checked and corrected the issues in the article based on your pdf, and you can read it again.

Reviewer 3 Report (Previous Reviewer 4)
Comments and Suggestions for Authors
Nice study--no further issues.
Author Response
No response
Reviewer 4 Report (Previous Reviewer 2)
Comments and Suggestions for Authors
I appreciate the effort and attention of the authors in addressing my suggestions.
Author Response
No response.
Round 2
Reviewer 1 Report (Previous Reviewer 1)
Comments and Suggestions for Authors
Unfortunately, I haven’t seen any improvement in the major issues since the last version. The authors failed to compare their results with the previous publication by Tang et al. and were unable to highlight the new findings in the updated manuscript. Unless Plants restrict the citation of references in other languages, it should not be a problem to cite the study in Chinese. The authors emphasized that alfalfa is “highly heterozygous and highly repetitive”. But this is why an allele-aware reference genome is more comprehensive and informative for studying alfalfa genetics, isn’t it? Since many results included in this manuscript were previously reported with more informative analysis, the current manuscript is not appropriate to be published.
Author Response
- Dear authors, Three reviewers and the academic editor have recommended acceptance of your manuscript. However, the first reviewer has rejected it, primarily based on a lack of novelty, as there is a very similar and more comprehensible study (Tang et al., 2023) previously published in a Chinese journal. I think that this reviewer's report should not be ignored, and I request that you include in your manuscript a reference and comments on this previous work, briefly comparing the two studies, explaining which of your results are a mere confirmation of previous data and highlighting the novel contributions of your work. For example, you should explain why your study is limited to 15 MsGPAT genes, whereas the Tang et al. paper apparently identifies 73 members of this gene family (according to Table 1). Once you introduce these minor modifications, the manuscript could be accepted for publication in Plants.
Response: Thank you for your suggestion. We have cited that Chinese reference(Tang et al. 2023) and added explanations for the feasibility and novelty of this manuscript in lines 112-127, and explained the reasons for the small number of GPAT family members in lines 307-315.
This manuscript is a resubmission of an earlier submission. The following is a list of the peer review reports and author responses from that submission.
Round 1
Reviewer 1 Report
Comments and Suggestions for Authors
In this manuscript, Ma et al. identified 15GPATs which might be involved in the response to abiotic stress in alfalfa by using BLASTP against Medicago sativa genome with GPATs of Arabidopsis as query sequences. Although the topic could be potentially helpful for future studies of alfalfa tolerance to abiotic stress, similar results were reported by Tang et al in 2023 using the same approach (DOI: 10.11733/j.issn.1007-0435.2023.09.005). A better version of chromosomal localizations of MsGPATs at allele-aware chromosome level was provided by Tang et al. using all 4 sets of chromosomes. Overall, this research falls short of novelty, and the manuscript was not well prepared. Some figures such as Figures 2, 3, 4 and 5, are incorrectly referred in the text, and the panel C in Figure 7 is missing. There are also a number of issues with the methods and analysis that need to be clarified in the next version.
- Line 159: This is not a novel finding. It has been well known that alfalfa is more closely related to Medicago truncatula and Glycine max. More informative conclusions should be inferred from the result.
- The synteny analysis between Medicago truncatula and Medicago sativa shown in Figure 3 is incorrect. Medicago truncatula has 8 chromosomes, but only 7 chromosomes were included for the analysis.
- The phylogenetic tree in Figure 4 is not well rooted, hence the trifurcation. This is probably why MsGPAT8 and MsGPAT13 fall into different groups.
- The authors state that MsGPATs in the same groups showed similar structures. However, the conserved motif in MsGPAT9 is obviously different from the other members in group III. What would be the explanation?
- Line 238: This is not correct. A weaker response should be slightly changed or no change of the gene expression after the treatment. It does not matter whether the gene is up or down regulated.
- The authors use published RNA-Seq data for analysis of the expression patterns of the identified MsGPATs, but the RNA-Seq data for tissue-specific expression analysis is generated from cultivar B47, not ZhongMu No.1, which is used for qPCR in this article. The expression pattern of a gene family could be significantly varied by cultivar.
- Time course expression of MsGPATs in leaf was measured under cold stress, which is fine. But for salt and drought treatments, the responsive MsGPATs and their expression patterns in roots rather than leaves would be more informative.
- There are 15 GPATs were identified in this study, but only 9 of them were used for testing the expression level with qPCR? What is the standard for choosing the 9 GPATs but not the other 6? The expression levels of some MsGPATs, such as MsGPAT5, MsGPAT8 and MsGPAT11 were not significantly changed under cold stress treatment. What does that mean? It needs to be further discussed which MsGPATs are involved in response to abiotic stresses and which ones are not based on the qPCR results.
Other issues
- The figures need to be reordered and should be numbered in the order they appear in the text.
- The species Latin scientific names should be italicized.
- For readers unfamiliar with plant responses to abiotic stresses, it would be helpful to include some background knowledge about ABA signaling in the Introduction part.
- In Figure 2, what is the difference between the dashed line and solid line in the outer cycle of the circular phylogenetic tree?
- The quality of Figure 4 is poor, and it is not necessary to divide Figure 4 into 3 panels.
- Figure 5 includes 2 panels, should be labeled separately.
- The blue bars (control) in Figure 7 are dispensable. It might be more straightforward by replacing all blue bars in each panel with a time point “0h”.
- Line 428: “specific expression” needs to be specified.
Comments on the Quality of English LanguageThe whole manuscript is generally readable, but grammar mistake and inaccurate word usage still exist in the current version. More editing is required.
Reviewer 2 Report
Comments and Suggestions for Authors
The results are interesting, the experiments and analyzes are adequately carried out and the conclusions are well supported. However, several aspects must be carefully attended to so that the manuscript reaches the quality necessary to be published.
It is necessary to correct the consecutive numbering of the Figures; The scientific names of the species must be written in italics and this also applies to the prefix cis. In Figure 1 it would be clearer to include the eight chromosomes and in Figure 7 it is necessary to include the salt response graphs.
Although the corresponding information is included in the materials and methods section, it is better to include the address of the websites consulted when they are mentioned in the text.
In section 2.5 it is necessary to explain why only 13 genes were included in the analysis and in section 2.6 to justify why the analysis was restricted to 9 genes. In this same regard, given that the analysis of expression in response to stress is the only experimental component of the report, it seems to me that it would be worth extending it to an analysis of response to ABA and/or tissue-specific expression that validates and enriches the information. rescued from the databases.

For me, that English is not my native language, the manuscript is understandable, however, some small editorial errors (highlighted in the attached document) must be addressed.
Reviewer 3 Report
Comments and Suggestions for Authors
Comments and Suggestions for Authors
Dear Author
It is my pleasure to review the manuscript entitled “Genome-Wide Characterization of GPAT Gene Family in Medicago sativa L. and Expression Analysis under Abiotic Stresses” a research article submitted to MDPI Journal, Plants. Authors of this manuscript identified and characterized a number of GPAT genes in M. sativa. They have performed physicochemical properties, phylogenetic relationships, gene structure, promoter cis-elements, and duplication events analysis of the GPAT genes using the alfalfa genome. Authors have also characterized expression patterns of those genes in tissue specific through a series of bioinformatic and lab experiments. Overall, the experiments, they performed, are well and the results are convincing. Thus, the presented results take up an important topic consistent with the profile of the Journal.
However, I have some suggestions, which might improve the manuscript to make important to the wider readers.
· Improvement in English is very necessary for clear understanding. Some sentences are not in correct structures. Many of them are with no meaning. Therefore, English checking by an expert will improve the quality of the article.
Introduction
Introduction should be more constructive with rationale of the study. Elaborate clearly, why this research is necessary. Introduction not adherence. Simultaneous and specific discussion is necessary
The author references various studies on GPAT TFs across different species. A brief discussion on gaps or limitations in the current understanding, especially related to M. sativa species, would strengthen the rationale for the present study.
End of the introduction, the objectives are stated, but emphasizing how this study adds new insights or approaches differently from existing research could highlight its novelty and importance more effectively.
Be careful on writing the scientific name
Results
isoelectric point (pI): use correct style
Line 119: ranged from from-----correct it
-kD or kDa??
Line 136; check all scientific names
Line 155; fig 5 or fig 3?????
Line 166; might be fig. 4b. And also, where is description of fig. 4a
Line 195; should be fig 5. Also make same style for fig. figure or fig.???
Line 215; Inàin
Line 217; Theàthe. Also many other places
I could not find fig. 7c for salt treatment
Materials methods
. Elaborate GFF, HMM at their first time use?
. Total RNA was extracted using an RNA Extraction Kit. Please elaborate on this process. Which method did you use for the RNA quantitation?
. What was the reference sequences for BLASTP?
Similarity percentage is 36%. It should be below 20%
Comments on the Quality of English Language
Extensive editing of English language required
Reviewer 4 Report
Comments and Suggestions for Authors
The manuscript describes a thorough and useful study. Please see comments on grammar and clarity improvements. Also, I appear to be missing Figure 7, part c, in my version of the manuscript.
Comments on the Quality of English LanguageThe grammar and clarity of presentation should be improved throughout including punctuation, subject-verb and tense agreements, and sentence structure. Cumbersome language can slow the reading and hamper understanding of the data. Also, the figures are a bit complex, so very detailed and clear summaries of them in the text would be excellent.